# Spatio-Temporal Synergy between Urban Built-Up Areas and Poverty Transformation in Tibet

**Yiting Su [1], Jing Li [1,*], Dongchuan Wang [2,*], Jiabao Yue [3] and Xingguang Yan [1]**

1   College of Geoscience and Surveying Engineering, China University of Mining and Technology-Beijing, Beijing 100083, China; taylorlarry960107@gmail.com (Y.S.); bqt2000204051@student.cumtb.edu.cn (X.Y.)
2   School of Geology and Geomatics, Tianjin Chengjian University, Tianjin 300384, China
3   College of Resource Environment and Tourism, Capital Normal University, Beijing 100048, China; yuejiabao2019@cnu.edu.cn
*   Correspondence: lijing@cumtb.edu.cn (J.L.); wangdongchuan@tcu.edu.cn (D.W.); Tel.: +86-13488887216 (J.L.); +86-13820085635 (D.W.)

**Abstract:** Understanding the causes of poverty and identifying the transformation characteristics of poverty is the basis for achieving poverty eradication. In order to clarify the availability of construction land for poverty assessment, this paper explores the spatio-temporal synergy between urban built-up areas and poverty transformation in Tibet. The following conclusions are drawn: (1) the built-up areas in Tibetan counties have been growing from 2013 to 2019; (2) the proportion of counties with very low and low levels of relative poverty have decreased significantly, and the overall spatial characteristics of poverty are "high in the center and low in the surroundings"; (3) the overall coupling-coordination level between the built-up areas and the relative poverty level is gradually improving from the initial antagonism, and the relative-poverty index shows a significant negative correlation with coupling coordination (correlation coefficient of −0.63); and (4) the built-up area has a strong explanatory power for the spatial distribution of regional relative-poverty transfer compared to temperature, precipitation, elevation, and slope. The results of the study prove that the built-up area cannot be directly used as an indicator factor when constructing the multidimensional relative-poverty model and, instead, should use urban built-up areas by region to participate in poverty-estimation models based on regional economic development.

**Keywords:** Tibet; urban built-up area; relative poverty; coupling coordination; spatio-temporal synergy

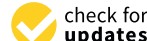



## 1. Introduction

Poverty is a long-term problem facing human society, and it is closely related to the economy, education, health care, the environment, and sustainable development. Understanding the causes of poverty and identifying the spatial characteristics of poverty are the basis for achieving poverty eradication [1–3]. The Chinese government has been committed to accelerating economic development for the benefit of the people, with poverty eradication as its key mission [4]. As the only contiguous and large-scale state-level extremely impoverished region in China, Tibet has achieved remarkable results in poverty eradication in recent years. Insisting on the combination of relocation and urban–rural integration is one of the strategies for poverty eradication in Tibet, which has injected momentum into the elimination of absolute poverty and solved the difficult problem of poverty eradication in Tibet. By the end of 2019, 74 previously poor counties in Tibet have been removed from the list. However, poverty is long-term and intergenerational in nature, so it is easy to get rid of poverty or return to poverty in a short period of time. The research group of Research on Tibetan Urban Poverty Population and Countermeasures explores the poverty characteristics of the Tibetan urban population from the perspectives of historical change, class, and family type, believing that there is a vicious cycle of poverty in Tibet [5–7]. Thus,



exploring the main factors affecting poverty and formulating effective policies are the keys to eliminating poverty.

The spatial characteristics of poverty eradication in different regions are also inconsistent, and many new features of the Tibetan poverty problem are the result of the interaction of complex factors such as the prominent manifestation of relative poverty, the unequal distribution of the poor masses and the advantaged masses in terms of structure and resources, market-consumption concepts, and the environment. Due to the specificity of Tibet's geographical location, the problem that "the local natural resources and environment cannot afford to feed its inhabitant" exists in the region, so the Chinese government takes "relocating poor people from inhospitable areas" as a key measure to implement the strategy of precise poverty alleviation [8]. The built-up areas of Tibetan towns and cities have been accelerated by the relocation policy, and their development status is closely related to the economy, environment, ecology, and human life in the region. Many scholars have explored the potential of using spatial, spectral, and textural characteristics of built-up areas, derived from remote-sensing imagery, to measure poverty. The research has shown the usefulness of built-up-area changes for monitoring local-poverty transfers, especially in exploring slums, the informal and formal settlements that are closely associated with poverty [9–12]. Different built-up areas have different unique spatial and textural characteristics (geometry, pattern, orientation, and spatial variability), so monitoring the area and spatial changes of them can be important for regional-development planning, risk assessment, and environmental nature management [13,14]. However, the small size of a region's built-up areas does not indicate that the living standards of the region's residents are poor, and it is unclear to what extent spatial and temporal variations in built-up areas affect the poverty levels of the region and whether there are regional specificities in this relationship. Clarifying the relationship between built-up areas and relative-poverty levels in Tibet is important for identifying and monitoring regional-poverty shifts.

The definition of poverty has shifted from the single dimension of economic and consumption level to multiple dimensions that encompass human development and the natural environment, which reflect the close and complex relationship between poverty eradication and other issues [15–17]. Up to now, there has been little research on the pattern of geographical differentiation of poverty at home and abroad, and most previous studies have focused on the causes of poverty from the economic and sociological perspectives, specifically from the perspectives of transaction costs, environmental changes, and natural-resource data [18–20]. As of 2019, the 18 poverty measures mainly rely on survey data, including related to income, consumption, health, education, and housing [21], but obtaining such survey data is time-consuming and labor-intensive, has poor data continuity, lacks spatial-distribution information, and is highly subjective, especially for the Tibet Autonomous Region, where comprehensive identification and analysis cannot be conducted [22]. Remote-sensing data have the advantages of being large-scale, with multiple temporal resolutions, to achieve long-term and large-scale studies at low cost. The rapid development of remote-sensing technology has provided an effective way to identify and monitor poor areas. How to use remote-sensing data to capture economic, income, health, and housing conditions to reflect poverty has become a hot topic of research [23–25]. Nighttime-light data have been widely used in the extraction of built-up areas. However, whether the built-up areas extracted from remote-sensing images can be used to explore poverty in Tibetan areas is a problem that has not been well addressed by the existing research. In order to help construct a multidimensional poverty model by remote sensing means realizing low-cost, large-scale, and long-time-series poverty remote-sensing monitoring afterwards, so this paper extracts built-up areas based on remote-sensing data combined with the Linear Spectral Mixture Model (LSMM) and the lighting-threshold method, and then explores the spatio-temporal-coordination effect of urban built-up areas and poverty transformation.

To explore the relationship between built-up areas and poverty, this paper selects Tibet as the study area by using remote-sensing images to extract built-up areas and then

carrying out correlation analysis with poverty transformation, to obtain the temporal and spatial synergy between the two. This article analyzes the spatial- and temporal-variation characteristics of the built-up area from 2013 to 2019, using a linear-regression-analysis model based on Landsat8 data and nighttime-light data, Annual VNL V1 of NPP-VIIRS. Subsequently, a multidimensional relative-poverty model was constructed and its spatial- and temporal-variation characteristics were analyzed using regression models. Finally, the spatio-temporal synergistic effects of urban built-up areas and poverty transformation are analyzed by coupled coordination-degree models and geographic probes.

## 2. Materials and Methods

### 2.1. Study Areas and Data

The Tibet Autonomous Region is located on the world's largest and highest plateau, in southwestern China, with an average altitude of over 4000 m and a total area of 1,202,200 km$^2$, known as "the roof of the world" and the "third pole of the earth". Due to the extremely fragile plateau ecosystem, the carrying capacity of Tibet for human activities is much lower than that of other regions. The complex and diverse topography, landscape, and ethnic culture have led to very scattered inhabitants and relatively backward infrastructure in most areas of Tibet, which greatly limit the sustainability of the livelihoods of Tibetan farmers and herders and make it difficult to consolidate the effect of poverty alleviation in the long run. As of 2015, the total population of Tibet is 3,239,700, and there are 74 national-level poverty-stricken counties, with the poor population mainly distributed in Changdu city, the Rikaze region, and the Naqu region [26]. According to the difficulty of information surveying, data processing, and the shaking off of poverty, 36 counties in five regions (Lasa, Rikaze, Naqu, Shannan, and Linzhi) that are out of poverty were selected as the study area, and the study period is 2013 to 2019. The spatial distribution is shown in Figure 1.

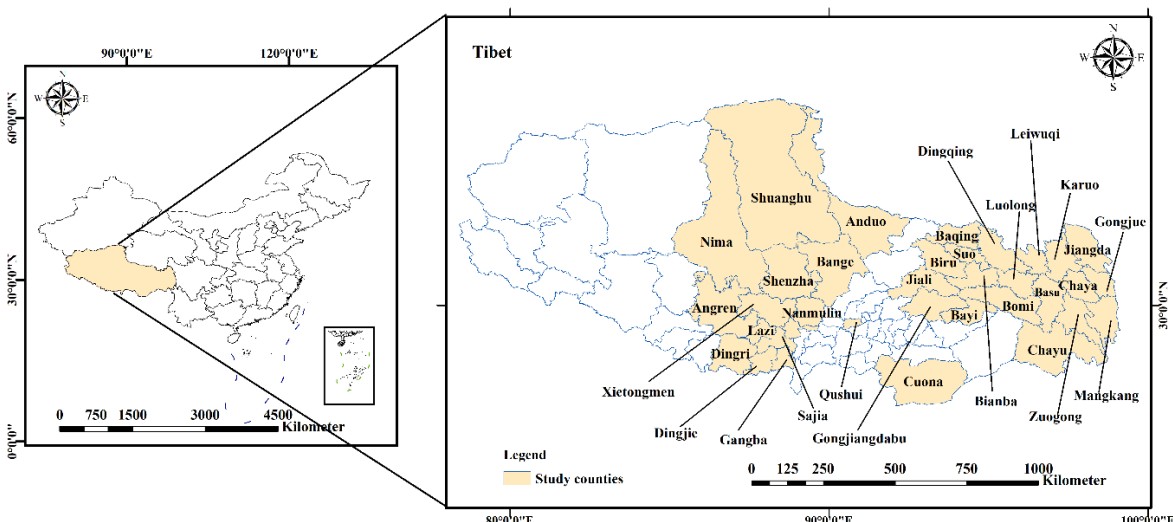

**Figure 1.** Study area: Tibet.

The research data are divided into two parts, statistical data are one part and remote-sensing data are the other part. The statistical data are obtained from the China County Statistical Yearbook, China Urban Construction Statistical Yearbook, Tibet Provincial Statistical Yearbook, Statistical Bulletin of Tibet Province, and annual statistics of counties in the CEInet statistics database. The remote-sensing data include nighttime-light data, named Annual VNL V1 of NPP-VIIRS andLandsat8 data, which were obtained from the National Oceanic and Atmospheric Administration (NOAA) and the United States Geological Survey (USGS), respectively. In addition, the time frame is from 2013 to 2019. The annual average nighttime-light data with a spatial resolution of 500 m is selected for this paper.

Landsat 8 data are with a spatial resolution of 30 m and a return period of 16 d, and the auxiliary-validation data are Google Earth high-resolution remote-sensing images with a spatial resolution of 1 m or 0.5 m. The data of county-level administrative-area divisions are obtained from the China Basic Geographic Information Data.

### 2.2. Research Methodology

In order to explore the spatio-temporal coordination effect of urban built-up area and poverty transfer, this paper has to extract the area of urban built-up area and construct the poverty index first. Therefore, the research methodology of this study is divided into three modules (Figure 2): (1) built-up-areas' extraction module (based on the combination of LSMM and nighttime-lighting-threshold method to extract spatial information of built-up areas in Tibetan towns); (2) multidimensional relative-poverty model-construction module (using Game Theory (GT) combined with Analytical Hierarchy Process (AHP) and Entropy Value Method (EVM) for subjective and objective assignment, adding time-series weighting method to construct multidimensional relative-poverty model); (3) analysis module (based on linear-regression models to analyze the spatial- and temporal-variation characteristics of urban built-up areas and relative poverty and to explore the spatial- and temporal-coordination effects of urban built-up areas and poverty transformation using coupling-coordination-degree model and geographical detector).

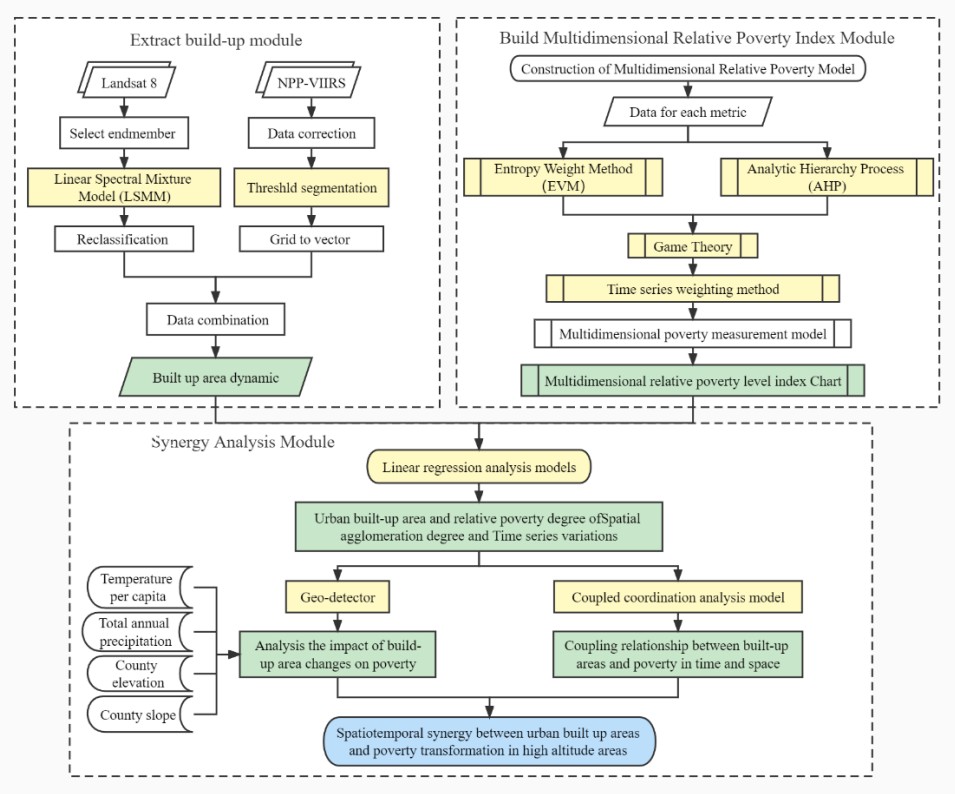

**Figure 2.** Flowchart of the proposed method.

### 2.2.1. Urban Built-Up-Areas Extraction Module

Nighttime-lighting data are widely used for the extraction of built-up areas, but there are problems of low resolution, 'light spillover', and susceptibility to scattering from road lights and water surfaces [27,28]. Building composition index (BCI), normalized difference impervious surface index (NDISI), etc., are more convenient to extract buildings, but bare soil and water bodies are confused with buildings, and the overall classification accuracy is not high [29–31]. LSMM solves the problem of mixed-image elements by obtaining the end-element components and can solve the problem of confusing water bodies with buildings. However, due to the similar composition of bare soil and building end elements

cannot be well distinguished [32]. This paper proposes to combine the LSMM with the nighttime-lighting-threshold method to extract the built-up area, which solves the problems of low resolution and confusion between bare soil and building end elements. The specific method is as follows:

(1)　Linear Spectral Mixture Model (LSMM)

LSMM can obtain information on the abundance of substances in multispectral or hyperspectral images based on the spectral characteristics of the substance, solving the problem of feature mixing [33,34]. LSMM assumes that the reflectivity of each pixel in the image is a linear combination of the reflectivity of each substance in the pixel or the endmember spectrum [35,36]. This is expressed as:

$$\begin{cases} R_{i\lambda} = \sum_{k=1}^{n} f_{ki} r_{k\lambda} + \xi_{i\lambda} \\ \sum_{k=1}^{m} f_{ki} = 1 \end{cases} \quad 0 \leq f_{ki} \leq 1 \tag{1}$$

where $R_{i\lambda}$ is the spectral reflectance of the i-th pixel in the $\lambda$ band; $r_{k\lambda}$ is the spectral reflectance of the k-th basic component in the λ band; $f_{ki}$ is the abundance of the k-th end element in the i-th pixel; $n$ is the number of end elements; and $\xi_{i\lambda}$ is the residual error value.

(2)　Nighttime-lighting-threshold method

The extraction of urban built-up areas with nighttime-lighting data focuses on obtaining the best threshold and segmenting the nighttime-lighting data with this threshold [37]. According to the accuracy, convenience, and automation of the method, the spatial-comparison method based on statistical data is selected in this paper, and the median interannual mean of the light values is taken as the threshold value for extraction, and the extracted area is corrected with the actual area until the difference between the two is minimized, at which point the threshold value is the best threshold value (Table 1).

**Table 1.** Nighttime-lighting-data thresholds.

| Year | Threshold |
|------|-----------|
| 2013 | 0.2569 |
| 2014 | 0.1525 |
| 2015 | 0.3289 |
| 2016 | 0.3992 |
| 2017 | 0.1874 |
| 2018 | 0.3556 |
| 2019 | 0.1935 |

LSMM was modified to make it suitable for Landsat data type. Matlab language was changed into Javascript language supported by Google Earth Engine (GEE) platform. The study area was divided into four land use categories: buildings, vegetation, water bodies and others. The distribution map of building coverage in the study area was obtained (taking the Sangzhuzi District of Rikaze as an example, Figure 3a). As shown in Figure 3a, the mixing of buildings and bare soil was serious. Therefore, the LSMM was combined with the Nighttime lighting threshold method to reduce the influence of bare soil on the extraction of buildings by using the characteristics of nighttime-lighting data. The results are shown in Figure 3b.

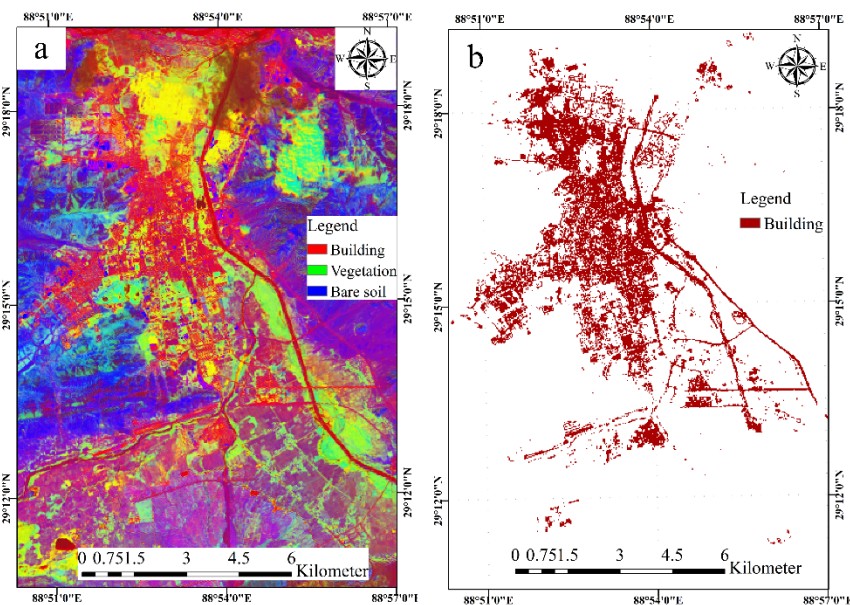

**Figure 3.** Map of built-up area in Sangzhuzi district, Rikaze. (**a**): building-coverage map calculated by the LSMM (red part); (**b**): building map extracted by the combination of the LSMM and the lighting-threshold method).

### 2.2.2. Multidimensional Relative-Poverty-Construction Module

The "Outlines of the China Rural Poverty Alleviation and Development Program (2011–2020)" requires scientificity, importance, comparability, typicality, data availability, and policy orientation for precise poverty alleviation [38,39]. The selection of the multidimensional poverty dimension refers to the multidimensional poverty-indicator system proposed by Wang Yanhui et al. [6,16,40] and the existing main factors (natural and socioeconomic factors) affecting poverty. Economic, social, and natural factors play a key role in the sustainable development of human and land resources [41], so select three dimensions: economic [18,42–44], social [6,16,42,43], and natural [6,45,46]. In addition, based on the research objectives and data accessibility, three dimensions were combined with regional poverty-reduction policies to construct a multidimensional relative-poverty-index system containing a total of 18 indicators in 8 vectors, as shown in Table 2.

**Table 2.** Multidimensional poverty-evaluation-index system.

| Dimension | Orientation | Indicator | Description | AHP Weights | EVM Weights | GT Weights |
|---|---|---|---|---|---|---|
| Economic dimension | Economic development | Per capita Gross Domestic Product (GDP) (RMB) | Reflecting the macroeconomic situation of the region [44] | 0.1461 | 0.0844 | 0.0439 |
| | | Residents' deposits (RMB) | Reflecting the economic sustainability of rural households [47] | 0.0962 | 0.0305 | 0.0603 |
| | Investment and consumption | Per capita local budget income (RMB) | Measuring the revenue capacity and level of government [18] | 0.0273 | 0.0366 | 0.0231 |
| | Industrial structure | Second industrial output (RMB) | Reflecting the economic income of the county's processing and manufacturing industry [42] | 0.0559 | 0.0461 | 0.1578 |
| | | Output of the tertiary industry (RMB) | Reflecting the income of the county's service economy | 0.0559 | 0.0426 | 0.0715 |
| | | Number of industries above scale (RMB) | The greater the number of factories, the greater the economic dynamics [43] | 0.0314 | 0.0414 | 0.0578 |

**Table 2.** *Cont.*

| Dimension | Orientation | Indicator | Description | AHP Weights | EVM Weights | GT Weights |
|---|---|---|---|---|---|---|
| Social dimension | Social security | Number of social service institutions (pcs) | Service coverage of the poor in favor of poverty reduction [16] | 0.0264 | 0.0552 | 0.0332 |
| | | Proportion of employed persons to total population (%) | Increasing the number of employed persons can improve people's livelihood [42] | 0.0264 | 0.0708 | 0.1100 |
| | | Number of fixed telephone users (person) | Number of durable goods reflecting the poor [16] | 0.0121 | 0.0308 | 0.0595 |
| | | Number of street offices (pcs) | Service security for the poor energy [43] | 0.0121 | 0.2612 | 0.0328 |
| | Infrastructure | Agricultural machinery power(w) | The higher the mechanical power, the lower the poverty level [48] | 0.0264 | 0.0381 | 0.0612 |
| | | Per capita facility agriculture area ($km^2$) | Increasing the area of facility agriculture and the efficiency of agricultural production [45] | 0.0664 | 0.0432 | 0.1018 |
| | Health and medical community | Number of beds per capita in health institutions (berth) | Reflecting the level of medical care [42] | 0.0264 | 0.0365 | 0.0525 |
| | Educational level | Number of primary and secondary school students (person) | Reflecting Education Resources [43] | 0.1320 | 0.0411 | 0.0433 |
| Natural dimension | Resource endowment | Per capita output of grain (kg) | The material resources of the population, which play a crucial role in the ability to withstand economic shocks at the population level [46] | 0.0629 | 0.0461 | 0.0862 |
| | | Per capita oil production (kg) | Same as above | 0.0629 | 0.0315 | 0.0439 |
| | | Per capita meat production (kg) | Same as above | 0.0629 | 0.0312 | 0.0603 |
| | | Total area of crop sowing per capita ($km^2$) | Same as above | 0.0629 | 0.0314 | 0.0231 |

To make the indicators comparable across different units, the extreme difference normalization was used to normalize the indicators of different dimensions, the formula is: $X = \frac{(x_{max}-x)}{(x_{max}-x_{min})}$, where $x$ is the value of each indicator. Each indicator of all counties in a fixed year is calculated according to the normalization formula and obtain the standardized values of 18 indicators from 2013 to 2019. The fixed-index values of all counties are in the range of 0–1, so the index values between counties are comparable, but the data between different years are lack of comparability. In order to solve this problem, indicators are assigned weights using a time series weighting approach to make them comparable.

The processed dimensionless indicators, are assigned to construct a multidimensional relative-poverty-index model. First, AHP and EVM are used to calculate the weights and combine the results of them using GT (Table 2), and then the normalized index value is weighted by time-series weight, so that the years are comparable (Table 3). Finally, the multidimensional relative-poverty model (Equation (10)) is used to multiply the processed indicators and GT weights to obtain the multidimensional relative-poverty index. In order to better demonstrate the spatial- and temporal-variation characteristics of relative poverty, K-means clustering analysis was used to classify relative poverty degree into five levels, according to the relevant literature [23,49,50]: non-poverty (0.0–0.1), slight poverty (0.1–0.2), mild poverty (0.2–0.4), moderate poverty (0.4–0.6), and severe poverty (0.6–1). The weights in Tables 2 and 3 are the example of Sangzhuzi district, and the calculation method is the same for the remaining 35 counties and districts. The specific process is divided into 5 steps.

**Table 3.** Time-series weights.

| Year | Time Series Weights |
|------|---------------------|
| 2013 | 0.2025 |
| 2014 | 0.1625 |
| 2015 | 0.1530 |
| 2016 | 0.2592 |
| 2017 | 0.2228 |
| 2018 | 0.1954 |
| 2019 | 0.2025 |

The first step is EVM, which is an analytical method to determine the weights of indicators based on the degree of variation of each indicator value [51,52]. Assuming that there are $m$ indicators and $n$ samples in the evaluation, where the contribution of the $i$-th indicator $A_i$ is under the $j$-th sample, $P_{ij}$ is shown in Equation (2):

$$P_{ij} = \frac{x_{ij}}{\sum_{i=1}^{m} x_{ij}} \tag{2}$$

Use $E_j$ to express the total contribution $E_j$ of all scenarios to attribute $X_j$, as shown in Equation (3):

$$E_j = \frac{-\sum_{i=1}^{m} P_{ij} \ln (P_{ij})}{\ln m} \tag{3}$$

In Equation (3), $E_j$ tends to 1 when the contribution of each program is for an attribute that converges; in particular, when all are equal, the weight of the attribute is 0. Therefore, the size of the difference in the attribute values determines the size of the weight coefficient.

The attribute weights $W_J$ are calculated, as in Equation (4):

$$W_J = \frac{1 - E_j}{\sum_{j=1}^{m} d_j} \tag{4}$$

The second step is the analytical hierarchy process (AHP), where the basic idea is to build a judgment matrix from the scalar values obtained by comparing the importance between two adjacent indicators in the sequence, so as to obtain the weights of each evaluation indicator of AHP. This paper uses a 1–9 scale to construct the judgment matrix of each layer [53].

The specific application process of AHP is as follows: first, construct a judgment matrix A (the scale is based on Table 4), according to the established hierarchy, and normalize the judgment matrix by columns to obtain $a_{ij}$, where $a_{ij} = a_{ij} / \sum_{k=1}^{n} a_{kj} (i, j = 1, 2, 3 \cdots n)$; then the normalized judgment matrix is summed by the row direction to obtain $w_i$, the vector $w_i$ is normalized according to the formula $W_i = w_i / \sum_{k=1}^{n} w_{ki} (i = 1, 2, 3 \cdots n)$ to obtain $W_i$, and $W_i$ is the AHP weight. Finally, in order to check the consistency of the matrix, the consistency index is needed to calculate $CI = \frac{\lambda_{max} - n}{n-1}$, where the largest eigenroot $\lambda_{max} = \sum_{j=1}^{n} \frac{(AW)_i}{nW_i}$. The closer the $CI$ is to 0, the higher the consistency. To determine priority of each variable, we need a judgment from an expert. The process of determining the weights of indicators takes the top-level indicators as an example (as shown in Table 5), and the weights of other layers are obtained in the same way.

**Table 4.** The scale of the judgment matrix and its meaning.

| Scale | Meaning |
|-------|---------|
| 1 | Both factors have the same importance when compared |
| 3 | The former is slightly more important than the latter, when compared to the two factors |

**Table 4.** *Cont.*

| Scale | Meaning |
|---|---|
| 5 | The former is significantly more important than the latter, when compared to the two factors |
| 7 | The former is more strongly important than the latter, when compared to the two factors |
| 9 | The former is more extremely important than the latter, when compared to the two factors |
| 2, 4, 6, 8 | The middle value of the above adjacent judgments |
| Countdown | If the ratio of the importance of factor $i$ to factor $j$ as $a_{ij}$ then the ratio of the importance of factor $j$ to factor $i$ is $a_{ji} = \dfrac{1}{a_{ij}}$ |

**Table 5.** Determination method of index AHP weight.

| Dimension | Economic | Social | Natural | Weights |
|---|---|---|---|---|
| Economic | 1 | 1 | 2 | 0.4126 |
| Social | 1 | 1 | 1 | 0.3275 |
| Natural | 1/2 | 1 | 1 | 0.2599 |

The third step is GT-combination assignment, which is a mathematical theory and method for studying phenomena with a struggle or competition nature [54]. The basic principle is as follows: assume that there are L methods of assigning weights to evaluation indicators, the corresponding set of basic weight vectors is $w_k = \left\{ w_{k1}, w_{k2}, \ldots, w_{kj} \right\}$, ($K = 1, 2, \ldots, L$) and the combination-weight coefficients are $\beta = \{\beta_1, \beta_2, \ldots, \beta_L\}$, if L weight vectors are arbitrarily linearly combined as:

$$W_j = \sum_{k=1}^{L} \beta_k W_{kj}{}^T \tag{5}$$

In order to seek consistency and compromise between different weights, the calculation takes the minimization of the deviation of $W_j$ and $W_{kj}$ as the goal and optimizes the L linear-weight-combination coefficients $\beta_k$ in Equation (5) to obtain the optimal weights $W_j$. According to the matrix differential property, the most optimized linear-equation system with the first-order-derivative condition is:

$$\begin{bmatrix} W_1 W_1{}^T & \cdots & W_1 W_L{}^T \\ \vdots & \ddots & \vdots \\ W_L W_1{}^T & \cdots & W_L W_L{}^T \end{bmatrix} \begin{bmatrix} \beta_1 \\ \vdots \\ \beta_L \end{bmatrix} = \begin{bmatrix} W_1 W_1{}^T \\ \vdots \\ W_L W_L{}^T \end{bmatrix} \tag{6}$$

Normalizing the optimal combination coefficient $\beta_k$ obtained by processing Equation (6), we get $\beta_k^* = \beta_k / \sum_{k=1}^{L} \beta_k$, and the corresponding $j$-indicator game-theoretic combination assignment of the combination weights is:

$$W_j^* = \sum_{k=1}^{L} \beta_k^* W_{kj}{}^T \tag{7}$$

where the value of $L$ is 2, $W_1$ is EVM weight, and $W_2$ is AHP weight.

The fourth step is to assign time-series weights, the socio-economic level of each city varies significantly in different years, so the introduction of time-factor weights can make the data comparable between years. The calculation is as follows:

$$GDP_t = \frac{gdp_t}{Gdp} \tag{8}$$

$$W_t = \frac{GDP_t}{\sum GDP_t} \tag{9}$$

where $Gdp$ is the total growth rate of Tibetan region from 2013 to 2019; $gdp_t$ is the annual $GDP$ growth rate; and the share of $GDP$ growth rate as time-weight $W_t$.

The fifth step is to obtain the relative -poverty index by multiplying the indicator weights derived from the game-theory-combination assignment with the indicator values.

$$F = \sum_i^n y_{ij} W_j^* \tag{10}$$

where $F$ is the relative-poverty index, the larger the $F$ value is, the higher the poverty degree. $y_{ij}$ is the value of the $j$ th indicator of county $i$; $W_j^*$ is the weight of the indicator; and $n$ is the number of county units.

2.2.3. Analysis Module

(1) Linear-regression model

The linear-regression model was used to analyze the spatial-temporal-variation characteristics of urban built-up areas and multidimensional relative poverty. The regression slope $K$ is calculated using the least squares method.

$$K = \frac{\sum_{i=1}^n \left(x_i - \frac{1}{n}\sum_{i=1}^n x_i\right)\left(y_i - \frac{1}{n}\sum_{i=1}^n y_i\right)}{\sum_{i=1}^n (x_i - \overline{x})^2} \tag{11}$$

where, $x$ is the time variable, $y$ is the dependent variable representing the built-up area or relative poverty, and $n$ is the study period. $K > 0$ represents an increasing trend; $K < 0$ represents a decreasing trend.

(2) Coupling-coordination-degree model

To analyze the degree of interaction between built-up areas and relative-poverty levels in poor counties, a coupling-coordination-degree model of urban built-up areas and relative-poverty levels is constructed based on the concept of capacity coupling in physics [55,56]. The coupling-coordination degree is used to analyze the level of coordinated development of things, which can characterize whether the two systems are mutually reinforcing or constraining each other at different levels [57,58]. For a better presentation of the results, this paper refers to the study of related scholars and is combined with the actual situation of this study [59–61], the middle-index-segmentation method is used to classify the degree of coupled and coordinated development into six classes: serious imbalance (0–0.2), moderate imbalance (0.2–0.4), mild imbalance (0.4–0.5), primary coordination (0.5–0.6), moderate coordination (0.6–0.8), and good coordination (0.8–1.0). To calculate the coupling-coordination degree, the coupling degree is first calculated with the following equation:

$$C = 2 * \left\{ \frac{f(x) \times f(y)}{[f(x) + f(y)]^2} \right\}^{\frac{1}{2}} \tag{12}$$

In Equation (12), $C$ is the coupling degree between the built-up area of the town and the relative-poverty level. The coupling-coordination degree is then presented with the following equation to define the system's overall-development level of them, the coupling-coordination degree is then introduced with the following equation:

$$D = \sqrt{C \times T} \tag{13}$$

$$T = \alpha \times f(x) + \beta \times f(y) \tag{14}$$

In Equations (13) and (14), $D$ is the coupled-coordinated-development degree, $T$ is the integrated-development index of built-up area and relative-poverty level, $\alpha$ and $\beta$ are defined as the weight values of built-up area and relative-poverty level, respectively, and added together equal 1. Since built-up area and relative-poverty level are two independent systems, take $\alpha = \beta = 0.5$, respectively.

(3)　Geographical detector

Geographical detector can explore the spatial heterogeneity of a single variable or detect whether the spatial distribution of two variables tends to be the same [62], and are widely used in regional spatial heterogeneity and the evolution of spatial patterns of geographic factors. The geographic detector has strong robustness and can identify the driving degree of the combination of two factors by q-value. Since the average altitude of Tibetan plateau is above 4000 m, and most of the deep-poverty areas are in areas with poor natural conditions and fragile ecological environment. Despite the previous adoption of various capital operation and anti-poverty measures, some of the population in the region still cannot get rid of poverty. Even if we ignore economic and social factors such as system, policy, education, and resources, the impact of natural geography on poverty is still a major problem that cannot be avoided. This paper selects the built-up area and four factors that can represent the physical geography to analysis the mutual driving effects of two factors on relative poverty as shown in Table 6. Referring to previous studies, the relative poverty is taken as the dependent variable (*Y*), while the independent variables are composed of the built-up area per capita (*X1*), temperature (*X2*), precipitation (*X3*), elevation (*X4*), and slope (*X5*).

**Table 6.** Selection of geographic-detector factors.

| Variables | Name | Meaning | Classification Criteria |
|---|---|---|---|
| Y | Relative poverty | The relative poverty of each county | |
| X1 | Built-up area per capita | Built-up area per capita of each county | |
| X2 | Temperature | The annual average temperature of each county | Classification according to the natural discontinuity taxonomy |
| X3 | Precipitation | The total annual precipitation of each county | |
| X4 | Elevation | The average elevation of each county based on the zoning statistics tool | |
| X5 | Slope | Calculation of slope from focus statistics | |

The degree of influence of different influencing factors on regional differences in poverty is measured by factor probes [63], and the driver of poverty heterogeneity *q* is introduced, and *q* is calculated as in Equation (15):

$$q = 1 - \frac{1}{n\partial^2} \sum_{h=1}^{L} n_h \partial_h^2 \tag{15}$$

where *L* is the stratification of the independent variable *X*, *n* is the total number of samples in the study area, and $n_h$ is the number of samples in stratum *h*; $\partial_h^2$ is the variance of stratum *h* poverty; $\partial^2$ is the variance of poverty in the study area, and the value range of *q* is [0, 1], but the closer the value is to 1, the greater the influence of the factor on the spatial differentiation of poverty; when the value of *q* is 0, the factor *X* is independent of poverty.

## 3. Results

### 3.1. Evaluation of the Spatial and Temporal Patterns of Urban Built-Up Area

The linear-regression model is used to regress the area of built-up areas in Tibetan cities and towns from 2013–2019 to derive the trend values (K) of the change in the area of built-up areas in Tibet between those years, as seen in Figure 4a. From Figure 4a, the results show that the built-up areas of counties in Tibet as a whole are growing (K > 0), and the top five counties with faster growth are Sagya county (1.7168), Gangba county (0.9984), Sangzhuzi district (0.8848), Bayi district (0.7512), and Karuo district (0.5302). The spatial distribution of Figure 4b shows that most of the cities with faster development are in the city of Rikaze (red part), but the built-up surrounding areas are developing more slowly (blue area). The cities of Changdu and Nagqu have developed slowly. On the whole, the growth trend of built-up area in the eastern region is more obvious and shows a trend of aggregation.

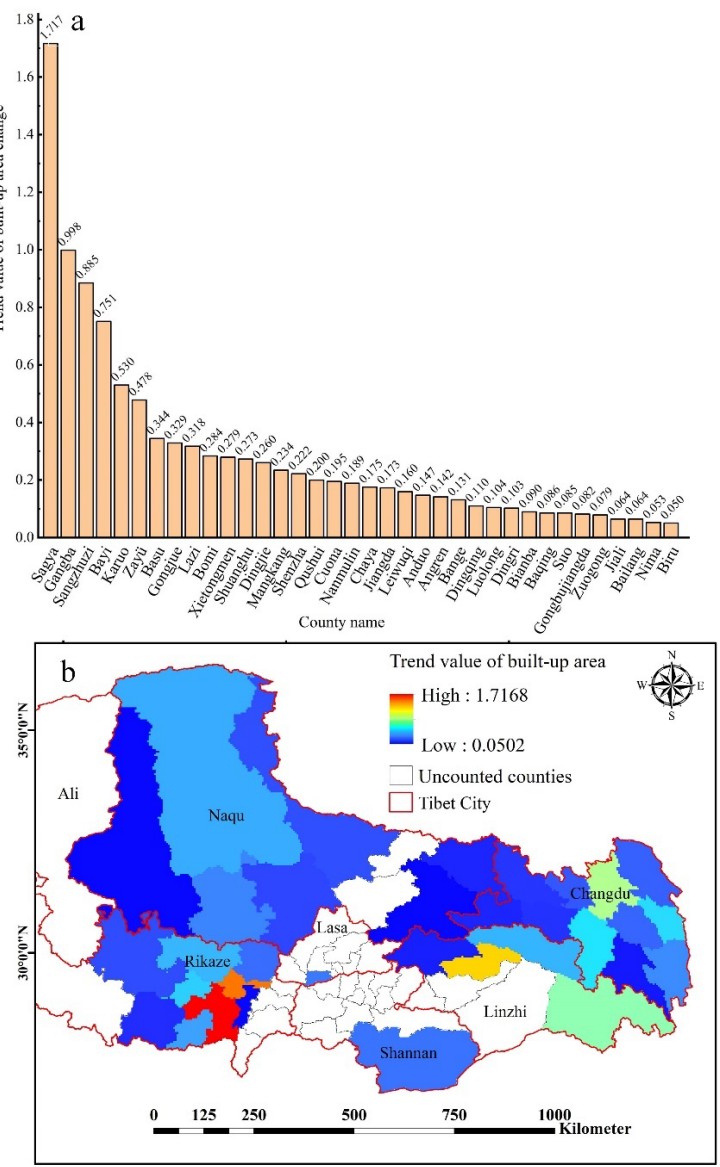

**Figure 4.** Trend values of built-up area from 2013 to 2019: (**a**) statistical map of county trend value; (**b**) distribution map of county trend value.

### 3.2. Spatial- and Temporal-Evolution Characteristics of Multidimensional Relative Poverty in Tibet

The temporal-evolution characteristics of the relative-poverty level in Tibetan counties are shown in Figure 5. Overall, the percentage of counties with severe poverty and moderate poverty significantly decreased between 2013 and 2019. The inter-annual trend value (K) is −0.20 in counties with severe poverty and −3.27 in counties with moderate poverty. The proportion of counties with mild poverty increased significantly (K = 3.08). The proportion of slight-poverty and non-poverty counties increased slightly, and the annual-variation-trend value is 0.1 and 0.3, respectively. These all indicate that the relative poverty level of Tibetan counties is increasing.

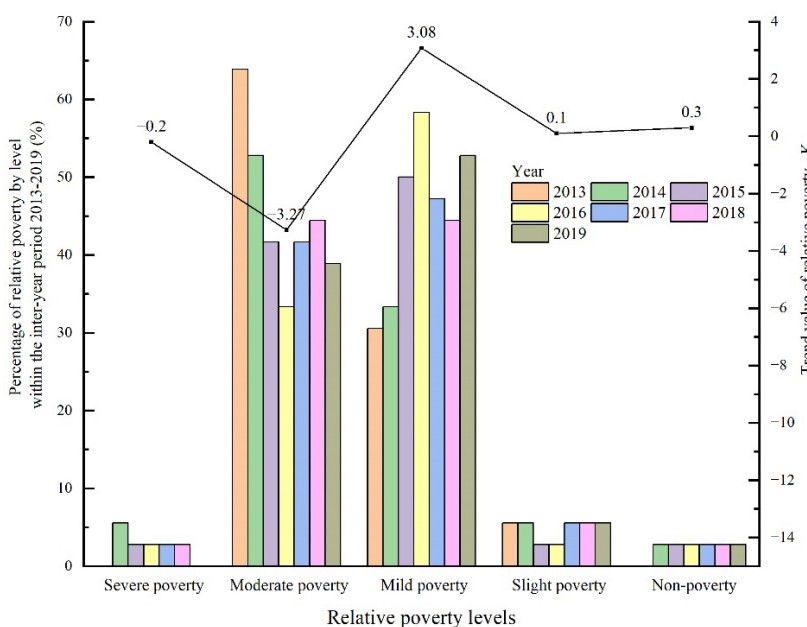

**Figure 5.** Percentage of relative poverty level within each year and the inter-annual change-trend value.

From the perspective of spatial divergence (Figure 6), there is a decreasing trend of poverty in adjacent counties of non-poverty counties, indicating that the counties with better economy can drive the development of the surrounding counties. In 2019, the counties in slight poverty and non-poverty include Sangzhuzi district and Bayi district, and no county is in severe poverty. However, there are still 38.89% of counties in moderate poverty, including Suo county, Leiwuqi county, Baqing county, etc. Besides, the relative-poverty level showd aggregation characteristics, with the central region showing mild poverty and slight poverty, while most of the surrounding counties show moderate poverty.

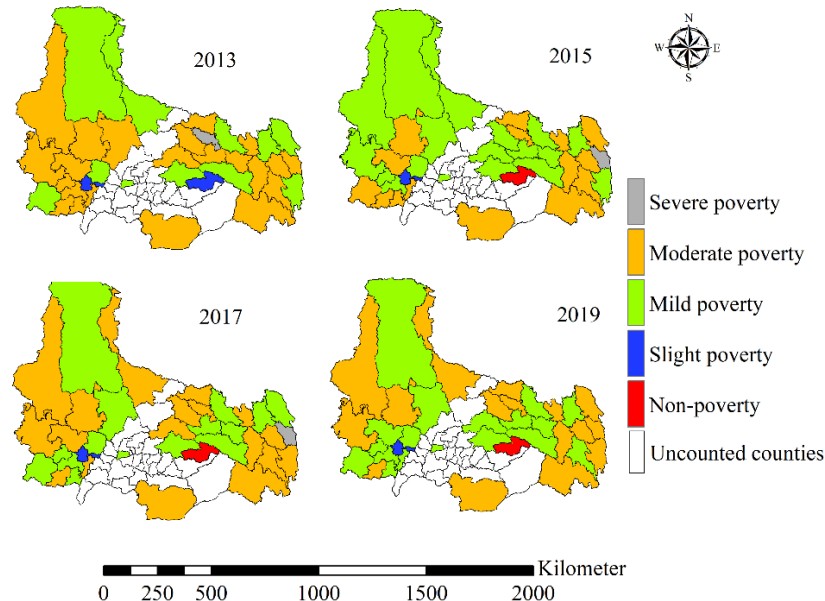

**Figure 6.** Spatial- and temporal-evolution characteristics of relative-poverty level in Tibet.

### 3.3. Spatio-Temporal Synergy between Urban Built-Up Area and Relative-Poverty Transformation

In this paper, the coupling-coordination model is used to calculate the coordinated-development degree of the two systems. From the temporal time dimension (Figure 7a), the median, mean, minimum, and maximum values of coupling coordination between urban built-up areas and relative poverty are increasing from 2013 to 2019. For instance, the minimum value rose from 0.10 to 0.22, and the percentage of seriously imbalanced counties is 0.00% from 2018 onwards. The proportion of mild imbalance showed a downward trend (the inter-annual trend value is −0.69). The proportion of primary coordination, moderate coordination, and good coordination showed an upward trend, with annual trend values of 0.69, 0.60, and 0.30, respectively, indicating that the overall level of coupling coordination in Tibetan counties is gradually moving from the initial antagonism toward improvement. By the end of 2019, the proportion of counties with primary coordination and above was 47.22%, but more than 50% counties were still faced with an imbalance between the urban built-up areas and relative-poverty levels (Figure 7b).

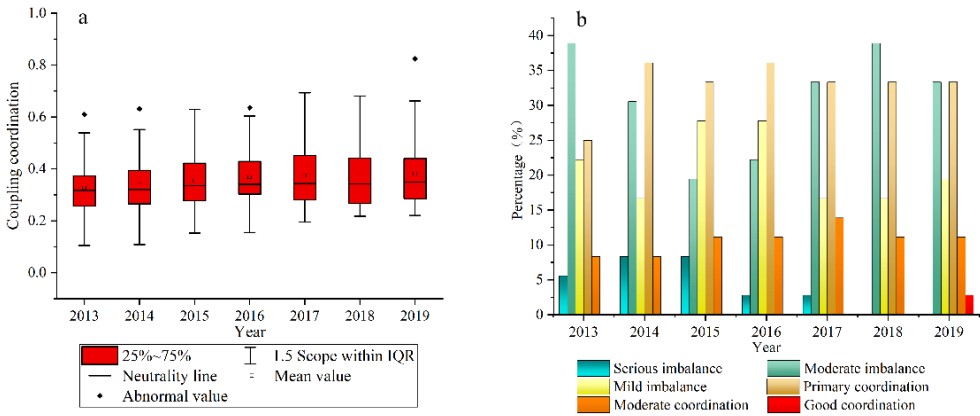

**Figure 7.** Temporal-evolution characteristics of the coupled coordination between urban built-up areas and relative poverty in Tibetan counties: (**a**) statistical value of coupling coordination; (**b**) percentage of each class during the year.

From the spatial dimension (Figure 8), the coupled coordination between urban built-up areas and relative poverty in the counties in southwestern Tibet is significantly better than in the counties of northeastern Tibet, among which the top-ranked counties are Gangba county, Sangzhuzi district, Bayi district, Dingjie county, and Sagya county, showing a trend of aggregation in the spatial location. The counties around Rikaze city have gradually shifted from a moderate imbalance towards the coordination level. However, the coupling coordination of Changdu city in the east is poor, and it has been in a state of imbalance, so the change is not obvious, including for Jiangda county, Mangkang county, and Zuogong county, which indicates that the area of urban built-up areas and the relative-poverty level in some counties are still in a low level of mutual restriction.

Comparing the poverty-level map (Figure 6) with the coupled-coordination map (Figure 8), the spatial-distribution characteristics are highly similar, where the poverty levels of counties in coordination are roughly for mild poverty, slight poverty, and non-poverty, so most of the counties in discoordination are in severe poverty and moderate poverty. Linear-regression analysis is used to explore the relationship between the relative-poverty index and the coupling-coordination degree. It can be seen from Figure 9 that the fitting curve shows a downward trend, and the correlation coefficient is −0.6281. Therefore, the relative-poverty index and the coupling-coordination degree are significantly negatively correlated. It shows that the poverty level and coupling coordination interact to some extent, and, overall, it shows that the poorer the region is, the worse the coupling coordination.

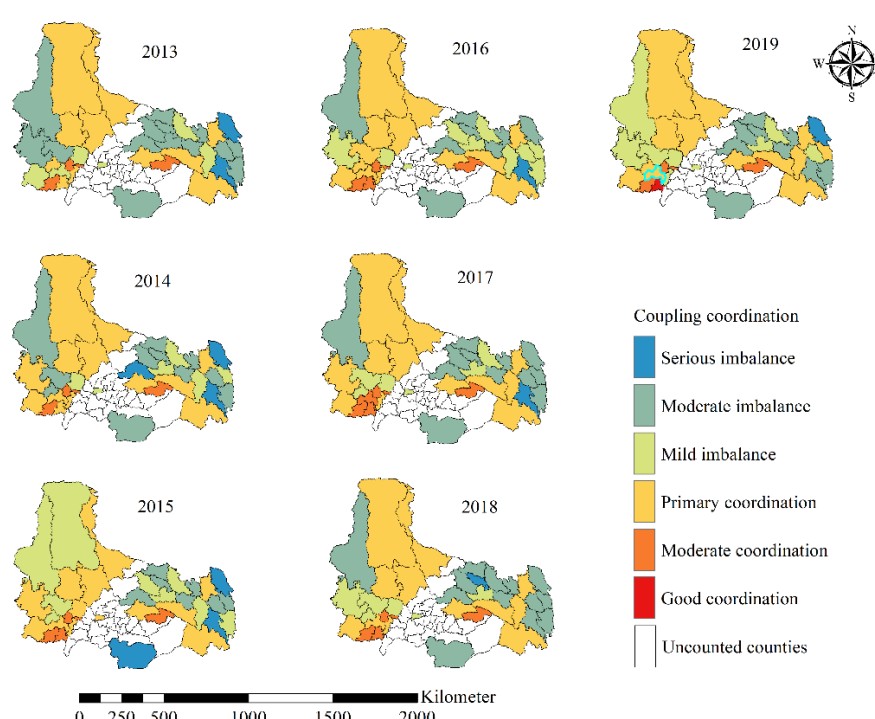

**Figure 8.** Spatial-evolution characteristics of the coupling coordination between urban built-up areas and relative poverty in Tibetan counties.

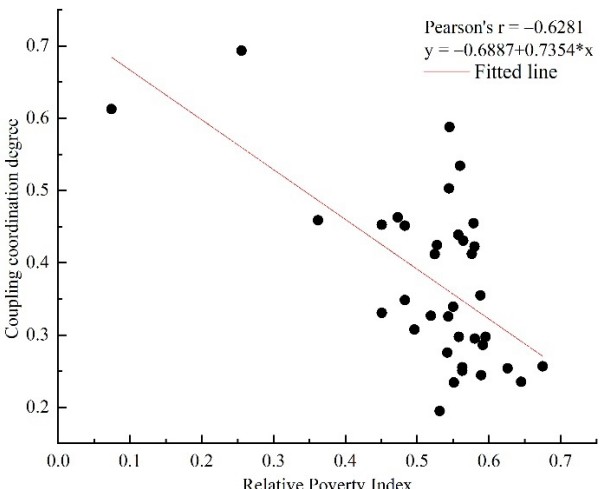

**Figure 9.** Scatterplot of relative-poverty index and coupling coordination.

*3.4. The Degree of Influence of the Association between Urban Built-up Area and Relative Poverty*

The q of the geographic detector is the explanatory power of the factor, and a larger q value represents the stronger explanatory power of the spatial distribution of the corresponding variable for that silver. From the perspective of a single factor, the per capita built-up area (q = 0.3103) is more powerful than the annual average temperature (q = 0.1207), the annual total precipitation (q = 0.0066), the average altitude (q = 0.0170), and the slope (q = 0.1221). From the perspective of double-factor interaction, the influence of double-factor interaction is significantly higher than that of a single factor. The combination of the built-up area (X1) and the remaining three factors (county average annual temperature X2, county total annual precipitation X3, and county slope X5) all influence regional relative poverty in a strong, synergistic manner (Table 7). Among them, the built-up area per capita and county average elevation (X1 ∩ X4, 0.5551), built-up area per capita and

slope (X1 ∩ X5, 0.5460), and built-up area per capita and precipitation (X1 ∩ X3, 0.5397). The combination of countywide average annual temperature and countywide total annual precipitation, as well as countywide average elevation and countywide slope, had little effect on regional relative poverty. Thus, the per capita building area has a strong explanatory power for the spatial distribution of regional relative-poverty transfer, compared to temperature, precipitation, elevation, and slope.

**Table 7.** Degree of synergy between explanatory variables and relative poverty.

| Interaction | Degree of the Relationship (q) |
| :---: | :---: |
| Temperature ∩ Precipitation | 0.2227 |
| Elevation ∩ Slope | 0.2440 |
| Built-up area per capita ∩ Temperature | 0.4870 |
| Built-up area per capita ∩ Precipitation | 0.5397 |
| Built-up area per capita ∩ Elevation | 0.5551 |
| Built-up area per capita ∩ Slope | 0.5460 |
| Precipitation ∩ Elevation | 0.2483 |
| Temperature ∩ Elevation | 0.2054 |
| Precipitation ∩ Slope | 0.0170 |
| Temperature ∩ Slope | 0.2328 |

## 4. Discussion

Poverty eradication is a common task of human society and a difficult problem facing China's social and economic development at present. The outline of "China Rural Poverty Alleviation and Development Program (2011–2020)" states that China will achieve the goal of building a moderately prosperous society by 2020 [64], with the focus on the central and western regions and the difficulty in the contiguous poverty-stricken zones. From 2013 to 2019, the socio-economic conditions in Tibet have significantly improved: there were 74 national poverty counties and 5369 poor villages in 2013, but the 628,000 people who have been documented in poverty are all out of poverty, so the incidence of poverty was reduced to zero by the end of 2019. As the only large-scale and contiguous poverty-stricken region in China, Tibet has made remarkable achievements in poverty eradication during these seven years, but there is still a lack of relevant studies on the Tibetan regions, and there is a lack of sufficient theoretical support to describe poverty identification and monitoring in the Tibetan regions, so this paper selects 2013–2019 as the research period to analyze the spatial- and temporal-response mechanisms of urban built-up areas to poverty transformation and discussion.

### 4.1. Combinatorial Weighting Method Based on Time-Series Weights

When constructing poverty models, most scholars utilize one or more methods for analysis. AHP is the most widely used, as it combines qualitative and quantitative analysis, but it is subjective and arbitrary [65,66]. EVM is more objective than AHP, but it ignores the opinions of decision makers [67]. The processing results obtained by the integrated grey evaluation method and data-envelopment analysis are susceptible to data extremes and require avoidance of linearity between input variables before use [68,69]. Game theory integrates the relationship between indicators, takes into account both subjective and objective weights, and enables the optimization of weights [67]. However, these related studies rarely involve long time-series poverty monitoring. The formation and development of poverty is a long-term process, so long time-series poverty monitoring can help to understand the causes of poverty and formulate corresponding policies to eliminate poverty. Based on this, this paper proposes to use long time-series data for analysis that can reflect the trajectory of poverty transfer, and the introduction of time series weights can enhance the comparability of data between years, which can extend the study of the problem from the static domain to the dynamic domain and make the results more reasonable. Therefore, it is of great practical significance to explore poverty-development

differences based on long time-series data to achieve regional-poverty eradication and sustainable development.

### 4.2. Comparison of Urban Built-Up Area Results and Product Data

How to extract the spatial information of built-up areas accurately and quickly has been a hot issue in remote sensing and urban planning. In this paper, LSMM and the light-threshold method are combined to extract the urban-building area. In order to verify the reliability of the results, we make a qualitative comparison between the results of this paper and a global 30 m impervious-surface-map (MSMT_IS30) product, which was obtained from the National Earth System Science Data Center, National Science and Technology Infrastructure of China (Figure 10). The overall accuracy is 95.10%, the a kappa coefficient is 0.90 for this product data [70]. Since the product data are only extracted from the impervious surface of the world in 2015, the urban built-up area of Sangzhuzi district in 2015 is selected to compare the results. It can be seen from the diagram that the abundance map of the urban built-up area calculated by us is basically consistent with the product data. In addition, compared with the product data (Figure 10b), the results of this paper (Figure 10a) are the percentage of single-pixel ($30 \times 30$ m$^2$) buildings, so the results are more in line with the actual situation. Since the focus of this paper is not to explore how to extract buildings more accurately, there is no quantitative evaluation of the accuracy of building extraction. The subsequent study will extract long time series of built-up areas of Tibetan towns, based on the current method combined with the multi-source data.

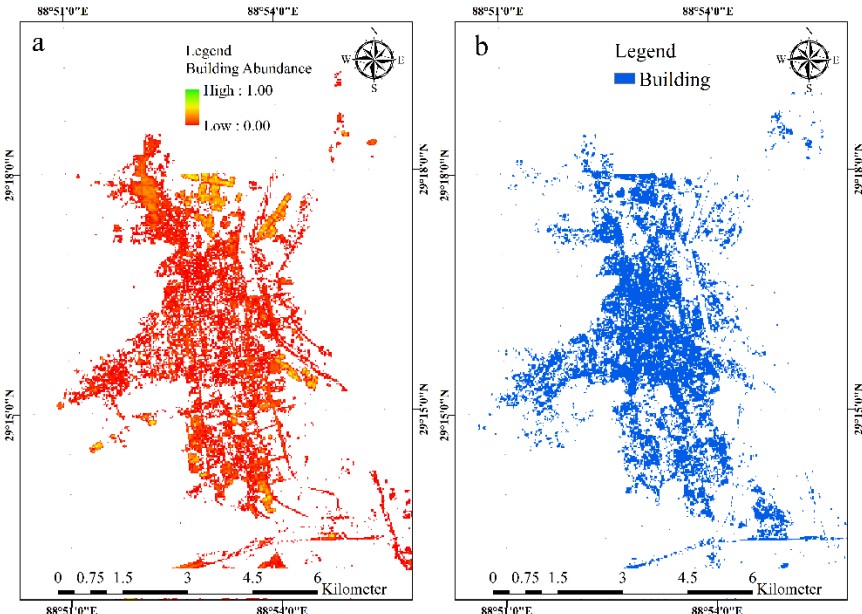

**Figure 10.** Comparison of the results of buildings in the town: (**a**) building area extracted by LSMM combined with lighting-threshold method; (**b**) global 30 m impervious-surface map.

### 4.3. Analysis of the Spatio-Temporal Synergy between Urban Built-Up Areas and Poverty Transformation

Regional poverty is inextricably linked to the development condition of county areas, due to ties between the economics, environment, ecology, and health. Urban built-up areas' extraction can spatially describe the traces of urban development, while built-up areas' growth is tied to regional population increase as well as corresponding changes in economic, social, and environmental factors [13,71]. Identifying changes in county built-up areas has important implications for poverty monitoring and identification. Previous studies have focused on the impact of commercial buildings and infrastructure on poverty. Yang et al. concluded that POI cost distance closely related to social and economic prosperity is one of the main factors leading to poverty in Chongqing, and that POI cost distance is negatively

correlated with poverty [17]. Ren et al. (2018a) showed that regional infrastructure has a significant impact on poverty in China, and the improvement of infrastructure is positively related to the living standard of residents [42]. However, few studies have explored the overall built-up area, housing conditions, road construction, etc., which can also affect people's living standards and economic status. It is unclear what the impact of the overall urban built-up area is on poverty, so the built-up area cannot be used directly as a factor to construct the poverty index. Based on this, this paper discusses the spatial and temporal-synergistic effect between urban built-up areas and poverty transformation.

By extracting the built-up area, the results show that the built-up area of Tibetan counties shows a continuous increase from 2013 to 2019. From 2016 to 2020, Tibet will implement a cumulative investment of RMB 15.95 billion in the construction of relocation projects to alleviate poverty, for the construction of resettlement houses and supporting the infrastructure in resettlement areas [8]. The implementation of the policy has accelerated the expansion of built-up areas in Tibetan regions. In addition, the proportion of counties with relatively severe poverty and moderate poverty in Tibet is decreasing year by year, and significant achievements have been made in the fight against poverty and historic elimination of absolute poverty [72].

The overall level of coupling coordination between urban built-up areas and relative poverty is gradually better from the initial opposition. However, there are still counties in uncoordinated state, which implies that the coordination phenomenon is not stable and may delay the effect due to urban construction. In addition to the fact that poverty is influenced by a variety of factors, it is closely related to economic growth, environmental protection, ecological restoration, and sustainable use of resources [48]. Analysis of the impact of built-up areas on poverty is by geographic detector, and we find that the influence of the interaction of two factors is significantly higher than that of the single factor, where the built-up area per capita and elevation are more influential than temperature and precipitation on the spatial distribution of regional poverty. Relevant scholars have discussed the main reasons for the differences in poverty in Tibet, including physical capital poverty and social capital poverty. The poverty caused by physical capital includes regional differences and natural conditions. Tibet has complex terrain, high altitude, inconvenient transportation, and a closed market, which lead to the lack of obvious regional advantages in development. Poverty caused by social capital includes urban development, rural-housing construction, industrial parks, and transportation facilities, which are the main factors driving the expansion of construction land. Industrial development and economic growth brought about by the expansion of good construction land have largely promoted the transformation of urban and rural development and poverty alleviation [12,73–75]. In addition, the inter-annual variation characteristics of built-up areas based on time-series data have a greater impact on poverty transformation than natural factors.

However, the corresponding relationship between the change in built-up area and poverty cannot be described simply by a linear relationship, and the coupling coordination between the built-up area of urban areas and the relative poverty level, in up to 50% of counties in Tibet by the end of 2019, is still in a state of disorder. Studies have also shown that the impact of land expansion in built-up areas on poverty may be dynamic [9–12]. By analyzing the spatial distribution relationship between relative poverty and the coupling coordination, we find that the relative-poverty index is significantly negatively correlated with the coupling coordination. The higher the relative-poverty level, the worse the coupling coordination of the corresponding counties. This shows that the expansion of county built-up areas with better development has a positive impact to some extent, while the rapid expansion of county-construction land with higher poverty has little impact on regional development.

### 4.4. Research Limitations and Future Work

Although regional built-up areas have a high impact on relative-poverty levels, poverty is a complex and diverse system involving multiple dimensions, including not only natural

and socio-economic development but also the willingness of farmers to struggle and the organizational and managerial capacity of the government; so, in this study, it is difficult to determine the extent of the specific impact of built-up areas on poverty relative compared to other factors. Since 2007, Tibet's fiscal spending efforts are much higher than the national average and are five percentage points more than in 2014. For Tibet, with a more backward economy and a single industrial structure, the finance is bound to increase the efforts to support regional development. However, based on the perspective of policy paradox, we believe that "policies such as national main functional areas and ecological barrier construction will reduce development opportunities in Tibet and lead to the formation of policy-based poverty, while compensation and subsidies can only play a short-term and temporary role". In order to prevent the return of poverty after being lifteed out of poverty, it is necessary to monitor poverty dynamics on a large scale using time-series data and develop innovative pathways. According to the conclusions of this paper, the next research plan is based on the county's economic development and thesub-regional county built-up area, as one factors to build the long-term sequence of Tibet's multidimensional poverty map.

## 5. Conclusions

This paper attempts to established the relationship between urban built-up areas obtained by remote-sensing images and regional poverty, to clarify the availability of built-up areas in poverty assessment. Taking Tibet as the study area, the change of the built-up areas was monitored from 2013 to 2019, and it was found that the built-up areas present an overall growth trend. Among them, the built-up area of Rikaze city has increased faster than Changdu city and Naqu city. Subsequently, the multidimensional relative-poverty index was constructed, and the result shows that the proportion of counties with severe poverty (trend value is $-0.2$) and moderate poverty (trend value is $-3.27$) decreased in the region. The poverty level shows the aggregation characteristics of a lower poverty level in the central counties and a higher poverty level in the surrounding counties. In order to have a contrast between urban built-up areas and relative poverty, coupling-coordination analysis was used. In general, the level of coupling coordination between urban built-up areas and relative poverty is gradually improving from the initial opposition, and the relative-poverty index has a significant negative correlation with the coupling coordination (the correlation coefficient is $-0.63$). The results of geographical detector also showed that the factor groups that had a significant effect on relative poverty were the built-up areas per capita and elevation ($q = 0.5551$), built-up areas per capita and slope ($q = 0.5460$), and built-up areas per capita and precipitation ($q = 0.5397$). The combination of temperature, precipitation, elevation, and slope have a small effect on regional relative poverty. The interannual variability characteristics of built-up areas ($q = 0.31$) have a greater impact than natural factors on poverty transformation. Although the built-up areas in Tibet have a significant impact on relative poverty, they cannot be directly used as an indicator factor when constructing a multidimensional relative poverty model. The subsequent study should construct the model by region, according to the economic-development status of the regions.

**Author Contributions:** Conceptualization, Y.S., D.W. and J.L.; methodology, Y.S.; software, Y.S.; validation, J.L., D.W. and J.Y.; formal analysis, Y.S.; investigation, Y.S. and D.W.; resources, Y.S.; data curation, J.L.; writing—original draft preparation, Y.S.; writing—review and editing, J.L., D.W. and J.Y.; visualization, Y.S. and X.Y.; supervision, J.L.; project administration, D.W.; funding acquisition, J.L. All authors have read and agreed to the published version of the manuscript.

**Funding:** This research was funded by the Second Tibetan Plateau of Scientific Expedition and Research Program (STEP), grant number 2019QZKK0608 and the Chinese Academy of Engineering Cooperation Project, grant number 2020SX8.

**Data Availability Statement:** Not applicable.

**Acknowledgments:** We are thankful to the editors and the anonymous reviewers for their valuable feedback. Acknowledgement for the data support from "National Earth System Science Data Center, National Science & Technology Infrastructure of China (http://www.geodata.cn 26 March 2022)".

**Conflicts of Interest:** The authors declare no conflict of interest.

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
