# Peer review of "Spatio-Temporal Synergy between Urban Built-Up Areas and Poverty Transformation in Tibet"

_sustainability, doi:10.3390/su14148773_

Round 1

Reviewer 1 Report

After reading the manuscript Spatio-temporal synergy between urban built-up areas and poverty transformation in Tibet”, I highlight next remarks:

·      The first three conclusions displayed in the Abstract lack a linkage between urban built areas and poverty.

·      Introduction presents most arguments predominantly customized to the Tibetan context, but background is required to connect urban built areas and poverty transformation in a broad context. An in-depth literature review comprising relevant references in the field should help to enhance this section that shows a weak thread, i.e., unrelated points such as nighttime lighting, BCI and NDISI indexes. Research aim was not defined and main conclusions were not summarized herein. 

·      Instead of the confusing Figure 2, a general overview of a tiered methodology is required to link each stage with methods and analysis proposed in a concise manner (current section covers 5 out of 17 pages). Rationale to connect nighttime lighting with urban built-up areas is lacking. It is unclear why the former represents accurately the latter. Criteria to determine components of the framework shown in Table 1 are unknown, please further develop them. IN the same vein, normalization and aggregation methods and levels of achievement  ( thresholds) were not defined either. Description of indicators is misleading, units of measurement are also necessary. Weighting method is unclear, AHP or which one? Are the 36 counties with data availability representative of Tibet to support scientific soundness of the study? Period under study is unspecified. 

·      Weightings of components included in Table 1 must be revealed as well 

·      Section 3 combines methodological and practical aspects which hinders reader´s understanding. Some methodological points like the consideration of buildings, vegetation, water bodies and others as the four categories of built-up areas should have been addressed in Section 2. Please define the parameter K in Table 2. Bases to build values and define levels of Table 2 are unknown. Grounds to determine the six categories proposed in 3.3.  were not disclosed. Criteria to select geographic detector factors in Table 4 were not provided. What is the reason of subsection 3.4?

·      Discussion must be based on comparing results with other studies in the field. 

·      Relevant conclusions rather than a summary of results, practical implications, limitation encountered in the research and future lines of work should be also addressed in the last section.

·      Miscellaneous comments. English grammar and style must be enhanced. Numbering of lines is needed to facilitate review process. Avoid the use of the first person rather than the third one, i.e. “our government” which government was referred? Authors are suggested to adapt the content of sections to specifications of the journal (https://www.mdpi.com/journal/sustainability/instructions). Acronyms must be fully defined at first appearance, i.e., LMMS. All data sources, i.e., subsection 2.1. must be properly cited and referenced. Visualization of Figure 2 is poor. Remove surplus and unnecessary equations. Check adequacy of titles for Tables/Figures. 

Reviewer 2 Report

I feel that the paper is well written. However, there are some points to be revised.

1. Introduction is too long. How about making a literature review section? And the literature review should be related to the index and materials.

2. How about changing tables into graphs? It will be helpful to understand your result easily.

Reviewer 3 Report

The article is well structured and presented, therefore it can be accepted in the current form.

In detail, the methodological part exposed the structured of the method, the description of the indexes involved and the multicriteria decision methods such as The Analytic Hierarchy Process (AHP).

Results and discussion are also in line with the aim of the paper. The integration of AHP, EMV and Game theory is a good way to match together subjective and objective weights. Results underline the importance of time series weights to define a multidimensional socio-economic development level model. The authors justify with their results the relevance of monitoring poverty dynamics on a large scale in a time-series and developing innovative pathways.

Concluding, their contribution tries to fulfil this gap, providing an alternative perspective and methodology to face the poverty in China.

Author Response

Dear reviewers:      

Thank you very much for reading our article in your busy schedule. Your comments affirm our current work and we are very pleased to receive your approval. Analyzing the spatial distribution characteristics and time series characteristics of poverty can better formulate corresponding policies. Combining related methods and taking the advantage of large-scale observation of remote sensing data, we established the link between urban built-up areas and poverty distribution. We hope our study can provide an idea for related scholars to better apply remote sensing technology to poverty research and provide a new perspective for poverty eradication. Thank you again for reviewing our articles!

Sincerely,

Yiting Su

Reviewer 4 Report

Тhe paper ‘Spatio-temporal synergy between urban built-up areas and poverty transformation in Tibet’ seeks to explore the trade-off synergy between urban built-up areas and poverty transformation in Tibet.

I think this paper is a nice piece of research, and it adds to our knowledge about the specific issue of the effect of bulti-up areas and poverty. The review of literature serves well to the purposes of the study. The empirical section is well designed and competently implemented.

I have one suggestion, which I hope will help to further improve the study: to include theoretical references that justifies indicators included in the analysis – table 1.

Reviewer 5 Report

The manuscript entitled "Spatio-temporal synergy between urban built-up areas and poverty transformation in Tibet” aimed to quantify the trade-off synergy be-tween urban built-up areas and poverty transformation in Tibet Autonomous Region, China

 The paper is interesting and it is well written and structured. The paper is suitable to be published in Sustainability after considering the few comments below.

1) Page 2, last line. The abbreviation “LSMM” must be written in full.

2) Page 4, first paragraph. The selected times series of the NPP-VIIRS and Landsat-8 must be define. I believe they are from 2013 to 2019.

3) Page, 4, first paragraph, you have to mention the name of the nighttime product which is “Annual VNL V1” and it was processed to screen out ephemeral lights and background.

4) Page 8, 3.1 Evaluation of the spatial and temporal patterns of urban built-up area. How did you validate the extracted building. I mean the overall accuracy with other product or ground truth data. You can validate your results with the global free Impervious Surface Area or any other data.  

 5) Page 13, 4. Discussion. You have to discuss the limitation of your method and the future work.

Round 2

Reviewer 1 Report

·      A background that connects urban built areas and poverty transformation in a broad context is required in the Introduction. Research aim was not defined,  lines 96 to 103 list instead tasks undertaken.

·      A general overview of a tiered methodology is necessary to link each stage with methods and analysis proposed in a concise manner (current section covers by 6 out of 19 pages). Criteria to determine components of the framework shown in Table 1 are still unknown. In the same vein, normalization and aggregation methods and levels of achievement (thresholds) were not defined either. How is it possible to integrate into a single value all distinct values of indicators with different units of measurement? It is unclear how AHP method was applied to get weighting factors. Weights of components included in Table 1 must be thus provided. 

·      Section 3 combines methodological and practical aspects which hinders reader ́s understanding. The former points should be located in Section 2. Method to calculate K is unknown. Grounds to determine the five levels in 3.2 and the six categories proposed in 3.3. are also unclear.

·      All data sources, i.e., subsection 2.1. must be properly cited and referenced by avoiding website links. 

Round 3

Reviewer 1 Report

·      Criteria to determine components of the framework shown in Table 2 are still unknown. Normalization cannot enable to aggregate into a single value all distinct values of indicators with different units of measurement, that is a critical conceptual mistake. For instance, Number of industries above scale (in yuan), number of social service institutions (in pcs) or proportion of employed persons to total population (%). It is unknown how AHP method was applied to get weighting factors. Data/process (i.e., survey) to feed AHP method were not disclosed.

·      Section 3 combines methodological and practical aspects which hinders reader ́s understanding and should be properly located in Section 2. Method to calculate K is unknown. 

Author Response

Thank you very much for your interest and comments on our manuscript.
Please see our response.

Round 4

Reviewer 1 Report

.